# Recent Advances of Anderson-Type Polyoxometalates as Catalysts Largely for Oxidative Transformations of Organic Molecules

**DOI:** 10.3390/molecules27165212

**Published:** 2022-08-16

**Authors:** Zheyu Wei, Jingjing Wang, Han Yu, Sheng Han, Yongge Wei

**Affiliations:** 1Department of Chemistry, Tsinghua University, Beijing 100062, China; 2School of Chemistry and Environmental Engineering, Shanghai Institute of Technology, Shanghai 201418, China

**Keywords:** polyoxometalate, Anderson structure, organic synthesis, molecular catalysis, green chemistry

## Abstract

Anderson-type ([XM_6_O_24_]^n−^) polyoxometalates (POMs) are a class of polymetallic-oxygen cluster inorganic compounds with special structures and properties. They have been paid extensive attention by researchers now, due to their chemical modification and designability, which have been widely applied in the fields of materials, catalysis and medicine. In contemporary years, the application of Anderson-type POMs in catalytic organic oxidation reaction has gradually shown great significance for the research of green catalytic process. In this paper, we investigate the application of Anderson-type POMs in organic synthesis reaction, and these works are summarized according to the different structure of POMs. This will provide a new strategy for further investigation of the catalytic application of Anderson-type POMs and the study of green catalysis.

## 1. Introduction

The Anderson type of polyoxometalates (abbreviated as POMs) are an important class of structures in oxygen-bridged polymetallic cluster compounds, which can be expressed as [XM_6_O_24_]^n−^ or [H_x_(XO_6_)M_6_O_18_]^n−^ (M=Mo or W); its central heteroatom X can be replaced with the majority of transition metal atoms like Fe, Co, Cu, Ni and Mn, etc. The stable configuration of Anderson-type POMs is a flat and circular structure (Figure 1a,b). There are three different types of oxygen atoms on the surface of Anderson-type POMs, with different coordination modes, so the reactivity of different sites on the surface is fairly different. Additionally, the class of protonation of the surface oxygen atoms of the POMs are also significantly varied, which can be divided into the following two types (Figure 1c): one non-protonation µ_3_-O on the surface of Anderson-type POMs called type A; the other possesses protonated µ_3_-O called type B. Provided that the valence state of the central atom is high (oxidation state > 4), Anderson-type POMs are also found to exist in a fold isomer configuration. This is similar to ammonium heptamolybdate (Figure 1d,e). However, for the central metal atoms in the low valence states, this structure requires organic ligand protection to keep steady (Figure 1f) [1,2,3,4,5].

The oxygen atoms of Anderson-type POMs are surrounded by six octahedrons around the central hetero-atoms by sharing an edge to form a common planar ring structure, otherwise a folded-twisted structure. Moreover, the central hetero-atoms are varied and modified easily, so the structure has a high coordination activity. Their structure and properties can be more multifarious by further modification.

Classical structures of POMs are mostly uniform particles with nano, otherwise sub-nano, size. Due to the octahedral connection and easily modified characteristics of Anderson-type POMs, this leads them to become a good sub-nano building unit. They can be used to design and synthesize a variety of special sizes and properties of organic–inorganic POMs compounds which show exceedingly important application value in materials, medicine, catalysis and other fields.

The previous research about Anderson-type POMs mainly attach great importance to the synthesis and preparation of new compounds and characterization, etc.; the Wang group, Wei group, Zhou group, Niu group and Kreb group have made an army of contributions in this field.

In 2014, Wei groups developed a liquid-phase diffusion single-crystal growth sample addition method and a special and sample addition tube [6]. The invention can significantly improve the working efficiency of the single-crystal cultivation, and the quality of the obtained single crystal by this method is also significantly improved. The Wei group designed a large number of different Anderson-type POMs and crystal of their organic derivatives via this approach (Figure 2). They afterward defined a series of structure of new Anderson-type POMs and the relevant derivatives [7,8,9,10,11,12,13] through single-crystal X-ray diffraction, infrared spectrum, NMR spectrum and liquid chromatography, which greatly promoted the development of synthesis Anderson-type POMs and structural chemistry.

With the structural development of Anderson-type POMs, a mounting number of researchers began to study the modification of POMs which controlled the application in the catalytic field [14,15,16,17,18,19]. These studies illustrate that Anderson-type POMs not only possess acid-based properties, but also excellent redox performance for the contributor that is present in central atoms. Owing to their structural stability, they can also be used as catalysts and applied in homogeneous and heterogeneous reactions, and even through phase-transfer catalyst or ionic liquid catalyst after appropriate chemical modification. Therefore, Anderson-type POMs are regarded as new sub-nano molecules with high catalytic activity. Nowadays, the research about Anderson-type POMs materials and their functions has gradually become a hot field in polyacid chemistry [20,21,22,23,24,25]. Nevertheless, the early work in the catalytic application of Anderson-type POMs mainly focused on desulfurization of oxidative and the treatment of industrial wastewater [26,27,28,29,30]. The applications in organic reaction are still the confronting great challenge.

It is universally acknowledged that progress of society depends on the development of organic synthetic chemistry. Reducing or avoiding producing harmful products is the question of organic synthesis reaction and the focus of “green” chemical research [31,32,33,34]. With the development of Anderson-type POMs in the catalytic reaction, the researchers found that the replacement of a traditional metal or non-metal catalyst is feasible with Anderson-type POMs or their derivatives [35,36,37,38,39,40,41,42,43,44,45,46,47,48,49,50,51]. This system not only can improve the catalytic efficiency, but also generate less by-product in the process. This is exactly an environmentally friendly, green and efficient catalytic system.

The variety of Anderson-type POMs catalysts have been reported. This review summarizes previous research studies based on recent literature. These works are roughly divided into two major categories according to the anion structure (Figure 3). For convenience of description, we define the relevant letter for the following categories: (1) The catalytic application of simple Anderson-type POMs (abbreviated as P); (2) Catalytic application of organic modified Anderson-type POMs derivatives (abbreviated as PO). In these two categories, the catalysts are further divided into two minor categories according to the cationic form of Anderson-type POMs: (1) Simple inorganic cations (such as NH_4_^+^.Na^+^.H^+^.K^+^, the following is abbreviated as [I]^+^); (2) Organic cations ([O]^+^).

## 2. The Application of Anderson-Type POMs in Catalytic Synthesis

### 2.1. Catalyzation of Anderson-Type POMs (P)

#### [I]^+^P

In 2016, Kadam and Gokavi investigated the dynamics and catalytic mechanism of the Anderson-type POMs, chromium (III) molybdate catalyst, which originally formed via the reaction between [O=Cr^V^(OH)_6_Mo_6_O_18_]^3−^ from the oxidation of H[Cr^III^(OH)_6_Mo_6_O_18_]^2−^ and hydrazide without free radicals in the course of the reaction process. This work provides good ideas and inspiration for oxidation hydrazide by Anderson-type POMs [35].

In 2017, the Wei group reported a simple, mild and efficient method of aerobic oxidation amine by inorganic ligand-loaded non-precious metal catalyst (NH)_4_)_n_[MMo_6_O_18_(OH)_6_] (M=Cu^2+^; Fe^3+^; Co^3+^; Ni^2+^; Zn^2+^, n = 3 or 4) in water at 100 °C via one step, demonstrating that the catalytic activity and selectivity can be significantly improved by transforming the central metal atoms. In the presence of Anderson-type POMs, with O_2_ as the unique oxidant, the catalytic oxidation of fundamental amine and secondary amines, and the coupling reaction of alcohol and amines can be achieved to generate the relevant imines. This new catalytic system provides a new method for catalytic oxidation reaction through inorganic ligand-supported metal catalysts [36].

Soon after, Wei and collaborators proposed a Cu-based Anderson-type POM, (NH_4_)_4_[Cu(OH)_6_Mo_6_O_18_] used for oxidation of carboxylic acid to aldehyde in water (Figure 4). This system used oxygen as the sole oxidant in the mild condition and was also suitable for various aldehyde derivatives with different functional groups [37]. In this catalytic process, the catalyst (NH_4_)_4_[Cu(OH)_6_Mo_6_O_18_] can be recycled at least six times and maintain high catalytic activity. This method of preparing for carboxylic acid is not only simple to operate, but also avoids the use of expensive and toxic raw materials. The versatility of this catalytic route makes it possible for industrial applications.

In 2018, the Wei Group used inorganic ligand-supported Fe-based Anderson-type POMs (NH)_4_)_3_[Fe(OH)_6_Mo_6_O_18_] to prepare imines by aerobic oxidation of aldehyde/ketone and amine oxidation coupling with oxygen as the sole oxidation [38]. The catalyst is comprised of an inexpensive system (NH_4_)_6_Mo_7_O_24_∙4H_2_O and Fe_2_(SO_4_)_3_. This system can be applied in broad substrates with low loss of catalytic activity, proving that this catalytic system has great potential in catalytic chemical transformation. Additionally, the stably inorganic skeleton for the catalyst provides good stability and is reusable in the course of the reaction process; for this reason, the catalytic oxidation of halide and amine can be easily improved to the gram level and has the potential to apply in industry.

After that, the Wei group published an inorganic ligand-supported iodine catalyst (NH_4_)_5_[IMo_6_O_24_] using an efficient method of aerobic alcohol oxidation. This system is compatible with multiple functional groups with high selectivity and good stability [39]. Based on experimental results, they proposed a preliminary mechanism of iodine catalyst (NH_4_)_5_[IMo_6_O_24_] for alcohol oxidation (Figure 5). This process is similar to the enzymatic oxidation reaction, which can be divided into two independent semi-reactions: (NH_4_)_5_[I^VII^Mo_6_^VI^O_24_] mediated alcohol oxidation reactions and dioxygen-coupled oxidation reaction with [I^V^Mo_6_^VI^O_24_]. For the iodo-molybdic acid catalyst by the inorganic ligands supported, two oxidizing equivalents required for oxidation are stored at the iodine center. Meanwhile, the addition of additives exerts a significant influence on the progress of the reaction, because the presence of Cl^−^ and Ac^−^ as an electron transfer medium promotes the electron transfer efficiency of (NH_4_)_5_[I^VII^Mo_6_^VI^O_24_]. This proposed mechanism is meaningful for the application of new Anderson-type POMs in organic reactions. This catalyst is a kind of efficient and mild inorganic ligand coordination catalytic system with high valence of iodine. The system has a wide range of substrate tolerance, good selectivity and recyclability, which avoids using the toxic organic ligands and toxic oxidant. Compared with the expensive organic iodine reagent, this system is convenient to apply in medicine, spices and food additives. Besides, the structure of (NH_4_)_5_[IMo_6_O_24_] also provides more insight on designing the new Anderson-type POMs by replacing the metal ion in the backbone.

Subsequently, Wei and co-worker prepared a Zn-based Anderson-type POM, (NH_4_)_4_[Zn(OH)_6_Mo_6_O_18_], which was used as a catalyst for oxide cross-coupling reaction of halide and amine, oxide self-coupling reaction of amine, and the halide oxide reaction [40]. In this catalytic system (Figure 6), they chose benzyl amine and benzyl chloride as model substrates for the oxidative cross-coupling reaction with generating N-benzylidenebenzylamine in 32% yield, 47% selectivity. By screening the solvent, acetonitrile showed the best yield and selectivity with 89% and 90%, respectively. Additionally, shortening and extending the reaction time also affected the yield. In addition, the optimal reaction temperature at 60 ℃ and 1.0 mol% was the optimal amount of catalyst. Although the reaction uses O_2_ as an oxygen source, relevant imines, aldehyde and ketone can still be efficiently prepared. The inorganic ligand-supported zinc-based Anderson-type POMs are not only easy to prepare and synthesize by the hydrothermal method, but also easy to recover due to the heterogeneity of the reaction. Compared with precious noble metal catalysts such as rhodium, ruthenium, and palladium, this catalyst avoids the use of toxic, air and water-sensitive organic ligands. So, Anderson-type POMs still have great potential as a heterogeneous catalyst in organic reactions.

In the same year, Sawant and co-worker investigated the application of Co-based Anderson-type POMs, Co^III^(OH)_6_Mo_6_O_18_]^3−^, in catalyzed oxidation of acetaminophen in the aqueous media when the pH values are 1 or 2 [41]. In this reaction, the electron transfer from neutral acetaminophen to anion, and then in the step of rate determination, the free radicals are further oxidized to N-acetylquinone imide as an intermediate, and what is more, hydrolyze to obtain benzoquinone and acetic acid. The experimental results reveal that the formation of a weak complex among the reactants promotes the reaction.

In 2019, the Wei group reported Cu-based Anderson-type POMs, (NH_4_)_4_[Cu(OH)_6_Mo_6_O_18_], which were used to study the aerobic oxidation of alcohol [42]. The classical transition metal complex catalytic system required complex organic ligands or nitroso radicals as auxiliary catalysts, and even strong oxidants. Nevertheless, when many organic compounds are in contact with these strong oxidants, it can easily result in exploding. Therefore, the traditional oxidation reaction of alcohol possesses complex process, high-price, severe and dangerous reaction conditions. Although, this catalyst is synthesized in water from cheap ammonium heptamolybdate and copper sulfate. The results showed that this catalytic system was suitable for various substrates in the oxidation of alcohol with excellent selectivity and activity. This method is not only safe and efficient, but also environmental. It has the potential to achieve industrial application. The experimental results show that the catalytic mechanism of the catalytic system is as follows (Figure 7) [42]: The two oxidation equivalents required for oxidation are not only stored in the copper center, but can be transferred to six marginalized MoO_6_ on the unit for (NH_4_)_4_[Cu(OH)_6_Mo_6_O_18_]. The Cu-based Anderson-type POMs react with alcohol to generate active substance A, while A and E, as a pair of isoforms, can interconvert, possibly due to the transfer of electrons to the inorganic ligands MoO_6_ via the intramolecular oxygen bridge Cu-O-Mo. Thus, one of the Mo cells goes from positive hexavalent to positive pentavalent. Anderson-type POMs A firstly activate molecular oxygen generation activity species B. Subsequently, species B hybridizes by the O=O bond to obtain the highly reactive metal oxygen species C as an active oxidant. During this catalytic reaction, the detection by MS indicates that water is involved in the reaction, producing hydrogen peroxide, so that the catalytic system is closer to the galactose oxidase reaction. Cu^II^ and the hydroxyl radical act together as a single-electron oxidant to be a two-electron alcohol oxidation reaction. In the reaction of galactose oxidase, a single center which is copper reacts with oxygen, producing hydrogen peroxide as a by-product. High active species C reacts with ethanol to generate intermediate D, and then H atoms are extracting to generate aldehyde and regenerate E. Finally, the intermediate E (Cu^II^) was re-oxidized by hydrogen peroxide to substance A (Cu^Ⅰ^).

At the same time, the Wei group also prepared Fe-based Anderson-type POMs, (NH_4_)_3_[Fe(OH)_6_Mo_6_O_18_] and applied them in olefination epoxidation reaction with 30% H_2_O_2_ as oxidant. This catalytic system does not require additional reductant or free radicals as initiators, and the experimental operation is simple, completely avoiding the use of expensive, toxic precious metal catalysts. Moreover, it does not need air/water sensitive and commercially unavailable organic ligands or tungsten POMs. (NH_4_)_3_[Fe(OH)_6_Mo_6_O_18_] can be obtained by one-step reaction in 100 °C water. This system has successfully converted the various aromatic and aliphatic alkenes into the corresponding epoxy compounds, with a good yield and selectivity [43]. Subsequently, Wei and collaborators further used this system in oxidative esterification reaction of various aldehydes with alcohol, and the corresponding esters were achieved under mild conditions with high yields, including several drug molecules and natural products [44].

After that, the Wei group and co-worker developed Cr-based Anderson-type POMs, (NH)_4_)_3_[CrMo_6_O_18_(OH)_6_]. The various primary and secondary amines, and even dual primary amine, were successfully converted into the corresponding formamides and methylated using this catalyst with methanol as a potential formylation reagent. Compared with the high-valence Cr-based catalyst including CrO_3_ and K_2_Cr_2_O_7,_ etc. This one with low-valent chromium catalyst is more effective, safe, and green environmental [45]. Wei then reported another work using (NH_4_)_3_[Fe(OH)_6_Mo_6_O_18_] catalyzed formic acid and amine coupling to generate formamide in mild conditions [46]. In the same year, the Xu group also developed Fe-based Anderson-type POMs, Na_3_Fe(OH)_6_Mo_6_O_18_·5H_2_O to prepare for 5-formyl-2-furanformic acid from 5-hydroxyl methylfuranic acid. The related results also further proved that the catalyst has high catalytic activity under an aerobic environment [47].

In 2020, Wei and co-worker proposed Co-based Anderson-type POMs (NH_4_)_3_[Co(OH)_6_Mo_6_O_18_]. In the presence of this catalyst, it is effective in achieving from alcohol to esters when KCl is used as an additive and 30% H_2_O_2_ as oxidant (Figure 8). As the important complexes in the fields of biology, medicine, and fine chemicals, esters are usually prepared by carboxylic acids or active derivatives (acyl chlorine and anhydride) reacted with alcohol. The Co-based Anderson-type POMs catalytic system is cheap, stable, safe and efficient and can successfully make alcohol to esters under mild conditions. They first explored the substrate range of benzyl alcohol and fatty alcohol oxidation coupled with methanol to generate methyl esters. The results demonstrate that various substituted benzyl alcohols as well as fatty alcohols were performed efficiently and highly selectively; the highest yield can reach 99%. Subsequently, on the basis of the above work, they further extended the catalytic system to the esterification reaction of benzyl alcohol and fatty alcohol with other alcohols, the corresponding ester products were also obtained. More importantly, the natural product octranactone can also be prepared by this method with 66% yield. In this catalytic system, the experiment shows that the additive can significantly enhance the selectivity and activity of the catalytic reaction system. The single-crystal X-ray diffraction test showed that the chloride ion in the additive linked with CoMo_6_ forming a supramolecular dimer 2 (CoMo_6_Cl) served as the key catalytic intermediate, which then achieves alcohol-to-ester conversion through nucleophilic addition and β-H elimination. The catalyst enables to achieve an oxidation coupling reaction of various alcohols (aromatic and aliphatic) under mild conditions and produce the corresponding esters with high yield [48]. This work shows that Anderson-type POMs, as an important class of single-metal ion inorganic molecule carriers, are an excellent and environmentally friendly catalyst with great potential applications in organic reactions. As a multifunctional catalyst, it changes the structure and properties of the central metal, so that Anderson-type POMs show excellent catalytic activity and extremely high catalytic efficiency in the green catalytic oxidation of alcohol.

### 2.2. Catalyzation of Organic Modified Anderson-Type POMs (PO)

#### 2.2.1. [I]^+^PO

In 2017, Wei and co-workers reported an organic ligand-stable β-isomer Anderson-type POMs: [NH_4_]*β*-{[H_3_NC(CH_2_O)_3_]_2_MnMo_6_O_18_}[49]. The sub-nano-cluster acts as a highly active catalyst applied in selectively catalyzed oxidation of cyclohexanone, cyclohexanol, or a mixture thereof. The results show that the special butterfly topology of Anderson-type POMs significantly improves the catalytic activity. Additionally, due to its high selectivity and high catalytic conversion efficiency, it can enable use for the green industrial synthesis of adipic acid.

#### 2.2.2. [O]^+^PO

In 2017, the Yu group developed the first organic modified hybrid Fe-based Anderson-type POMs [N (C_4_H_9_)_4_]_3_[FeMo_6_O_18_(OH)_3_{(OCH_2_)_3_CNH_2_}] for the aerobic oxidation of aldehydes to acids in water (Figure 9) [50] with O_2_ (1 atm). They found that the alkalinity of additives played a great impact on yield. Benzaldehyde, which was used as substrate, reacted with different additives at 50 °C. When Na_2_CO_3_ (pK_b_ = 3.67) was used as additive, the yield of benzoic acid was 95%. When NaHCO_3_ (pK_b_ = 7.95) was added in the system, the yield of benzoic acid decreased to 63%. However, the additive replaced by Na_2_SO_3_ (pK_b_ = 6.8) the yield reduced to 26%. While Na_2_SO_4_(pK_b_ = 12.0) was chosen as additive, the lowest yield of benzoic acid was 5%. For organic alkaline additives, such as CH_3_COONa and Et_3_N, the yield of benzoic acid was 83% and 89%, respectively. Regarding additives containing neutral salts, such as NaBr, KCl, and NaCl, the desired product was obtained in moderate yield. The acid additive, NH_4_Cl, avoided the process reaction. These results indicate that the additive to Fe-based Anderson-type POMs, Fe^III^Mo_6_, has a great effect on the activity of the catalyst, perhaps because of the high tunability of the acid-based properties of POMs. Subsequently, they also studied the effect of the catalyst dosage and the reaction temperature on catalytic activity. Changing the amount of the catalyst has little effect on the yield of the product. The yield of benzoic acid reached the maximum at 50 °C, but increasing or decreasing the temperature showed the different result. When the reaction temperature increased to 70 °C, the yield decreases to 96%, possibly due to reduced contact surface between the oxygen (gas phase) and the catalyst. When the reaction was performed in a nitrogen sphere, the yield of the obtained products was 5% less. As a control experiment, the experiments conducted without Fe^III^Mo_6_, only a small amount of product generation can be detected. This Fe-based Anderson-type POMs system was carried out with oxygen as the sole oxidant under extremely mild aqueous solution conditions, and suitable to a variety of functional group aldehydes. This method is environmentally friendly, has a low cost and high catalytic efficiency, as well as the potential application in industry.

The Wei group and co-worker also found that halogen ions (X^−^) can be incorporated with sub-nanoscale organoalkoxyl ligands-modified Al-based Anderson-type POMs([(n-C)_4_H_9_)_4_N]_3_{AlMo_6_O_18_(OH)_3_[(OCH_2_)_3_CCH_3_]}) in solution, forming the stable supramolecular complex with the binding constant K = 1.53 × 10^3^. This system was used in oxidation to aldehydes from alcohol. Crystal structure analysis demonstrated this behavior that binding occurs between the halogen ion X^−^ and the unmodified three hydroxyl groups on the surface of {AlMo_6_O_18_(OH)_3_[(OCH_2_)_3_CCH_3_]}^3−^, forming multiple X···H-O [51]. The interaction in this supramolecular sub-nano-cluster system means that its catalytic activity for oxidation of aldehydes can be adjusted by introduction of halogen–halogen ions and water. Chlorine ions inhibit by blocking the active center of the cluster, and the catalytic activity of the cluster can be reactivated by replacing the chloride with water supra-molecules. The result shows that the presence of multiple synergistic hydrogen bonds is key to overcome the electrostatic repulsion between halogen ions and Al-based Anderson-type POMs. This work not only enriches the supramolecular chemistry of Anderson-type POMs, but also provides a new way of selective catalysts for oxidation reactions.

## 3. Summary and Outlook

In order to solve the environmental crisis brought by the traditional production of chemicals, the research and development of green chemical technology had gradually become one of the hot fields in chemical research since the 1990s. The use of chemical technology and methods to reduce or eliminate the production of substances that are detrimental to human health and the ecological environment is one of the research priorities in the green industry of chemicals.

Advances and innovations in chemical technology are often driven by new catalytic materials and new catalytic technologies. In the research in the catalytic field, as a new type of high-efficiency, green, cheap and safe catalyst for Anderson-type POMs, they possess excellent redox activity compared with other inorganic acids. At the same time, it has measure and controlled acidity, as well as excellent dual-functional properties in the catalytic reactions. Moreover, most POMs possess good solvability. Apart from that, they not only have a definite structure and size, but also can be further modified by organic group, giving them more excellent features. In addition, Anderson-type POMs as a catalyst have more selectivity, few side reactions and retard the corrosion for the equipment. Therefore, the catalytic application of Anderson-type POMs has a great prospect and research value in scientific exploration and green chemical technology (Figure 10).

In combination with relevant literature, due to the instability of Anderson-type POMs, sometimes it is necessary to perform a series of organic–inorganic hybrid modifications for the parent POMs. The simple Anderson-type POMs are widely used in organic synthesis reactions, and the organic group modified POMs derivatives can perform relevant structural modification for specific catalytic reactions. It can be seen that the difference of the molecular structure of POMs has a certain influence on the specific catalytic reaction, especially in the aspects of catalytic efficiency, selectivity and yield. The POMs modified by different inorganic cations or organic ligands have their specific characteristics functional, which enlightens us. We can modify and design POMs catalysts for special organic catalytic reactions, and have achieved the desired effect. Anderson-type POMs displayed remarkable catalytic characteristics in oxidation of alcohol. They also have extremely high efficiency in the formation of C-N, C-O and C-C bonds. By comparison to the traditional precious metal catalyst, Anderson-type POMs are not only easy to synthesize, but also have high catalytic activity and recyclability. It can be expected that this catalytic system will also play the significant role in the selective oxidation of hydrocarbons and will have overwhelmingly wide application in the field of industrial catalysis.

This paper aims at paying much attention to investigate Anderson-type POMs in organic reactions and provide some new strategies and ideas for researchers. It is believed that, in the near future, a mounting number of Anderson-type POMs will be used in organic reactions, which will promote the development of organic synthetic chemistry.

## Figures and Tables

**Figure 1 molecules-27-05212-f001:**
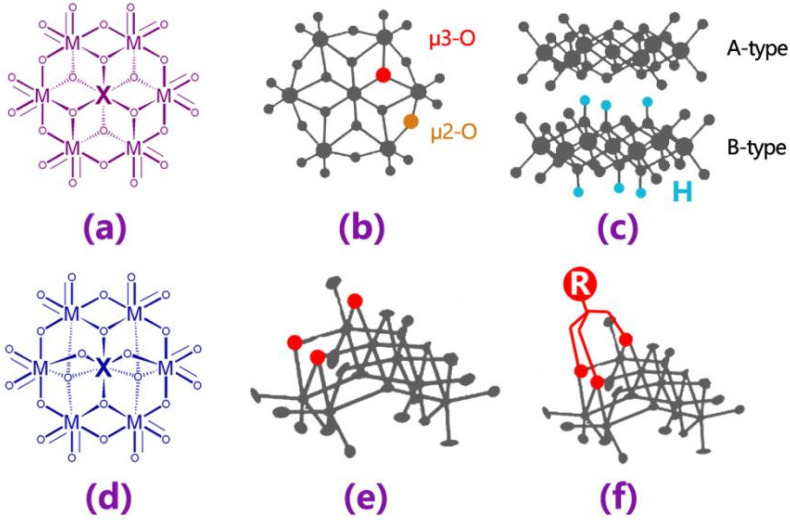
(**a**) Structure diagram of α-Anderson-type POMs; (**b**,**c**) Ball and stick model of α-Anderson-type POMs; (**d**) Structure diagram of β-Anderson-type POMs; (**e**) Ball and stick model of β-Anderson-type POMs; (**f**) Organic modified β-Anderson-type POMs.

**Figure 2 molecules-27-05212-f002:**
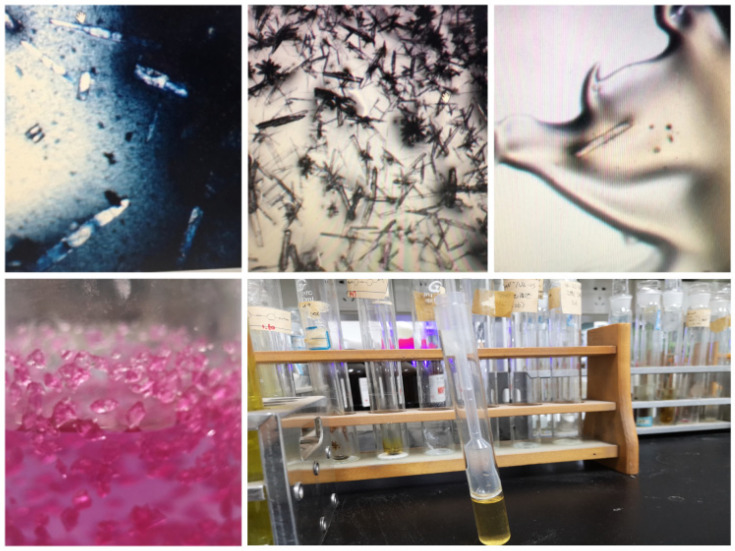
Several Anderson-type POMs crystals prepared by the special sample tube for liquid-phase diffusion technique [6,7,8,9,10,11,12,13].

**Figure 3 molecules-27-05212-f003:**
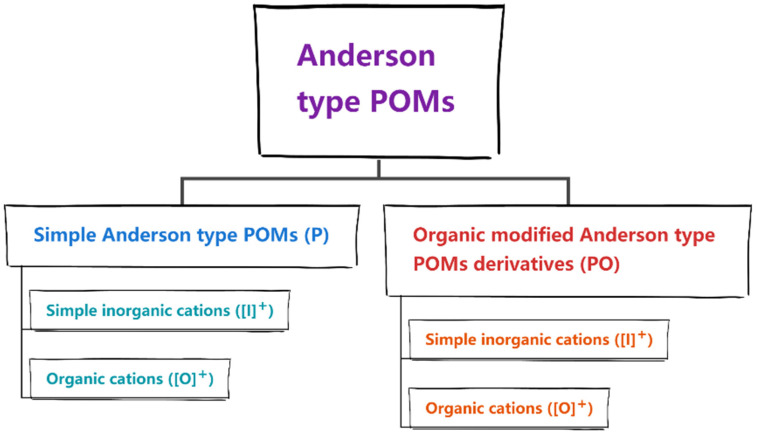
Tree-structure classification of Anderson-type POMs in this paper.

**Figure 4 molecules-27-05212-f004:**
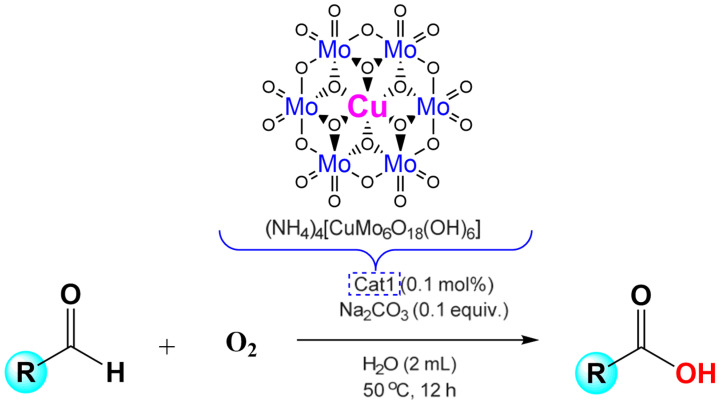
A method for preparing carboxylic acids by oxidizing aldehydes in water using (NH_4_)_4_[Cu(OH)_6_Mo_6_O_18_].

**Figure 5 molecules-27-05212-f005:**
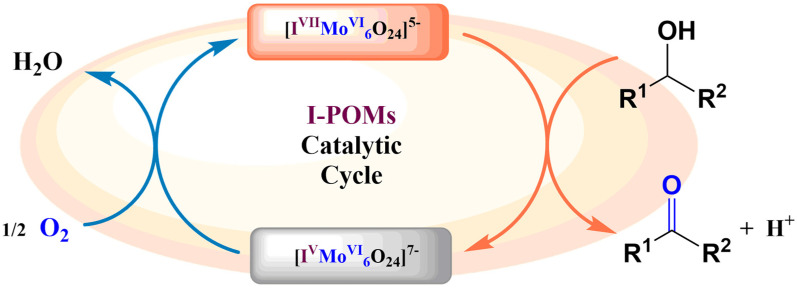
Proposed mechanism for the I-POM-catalyzed oxidation of alcohols.

**Figure 6 molecules-27-05212-f006:**
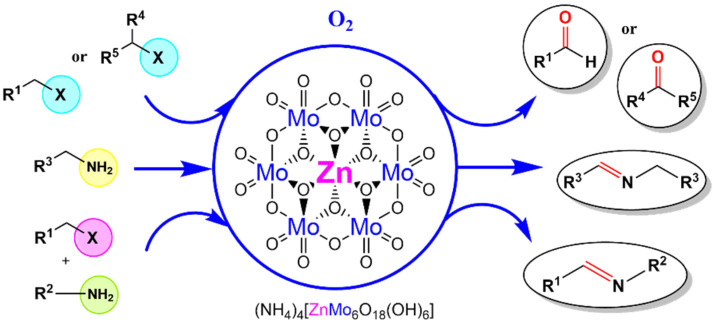
Zn-based Anderson-type POMs catalyze the oxidation of halides to aldehydes or ketones, and amines to sub-units.

**Figure 7 molecules-27-05212-f007:**
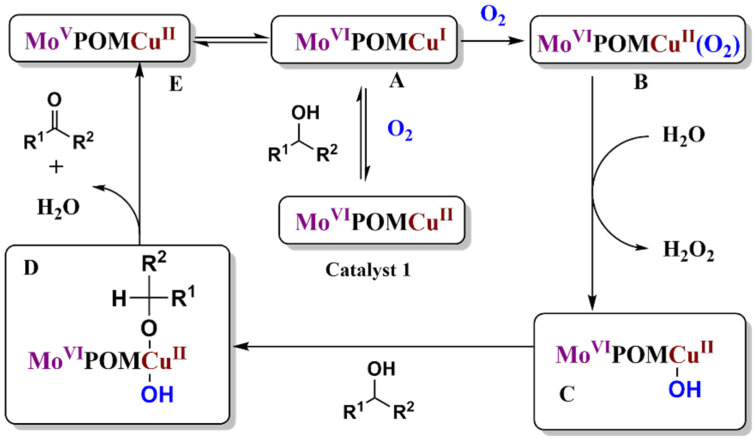
The catalytic mechanism of alcohol oxidation by using (NH_4_)_4_[Cu(OH)_6_Mo_6_O_18_].

**Figure 8 molecules-27-05212-f008:**
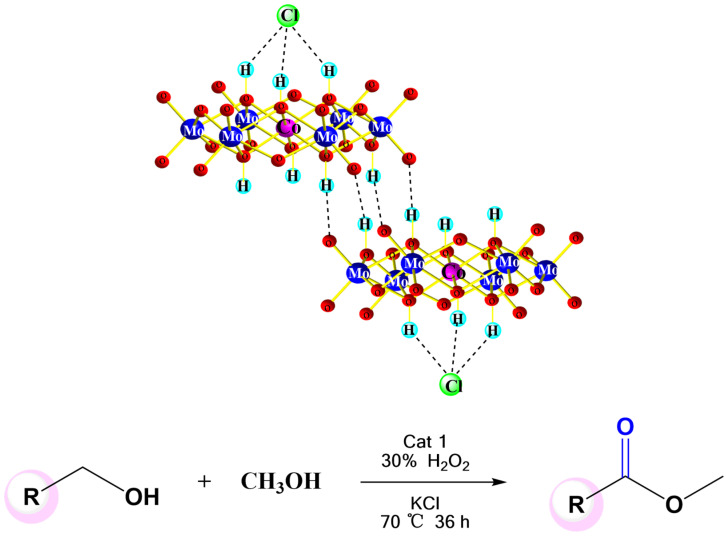
The supramolecular structure of dimer 2(CoMo_6_Cl) and its catalytic route.

**Figure 9 molecules-27-05212-f009:**
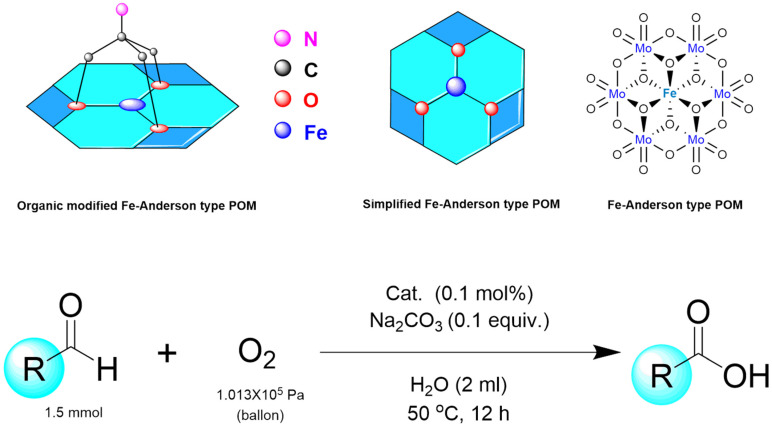
Catalyst anion structure and the catalytic reaction route.

**Figure 10 molecules-27-05212-f010:**
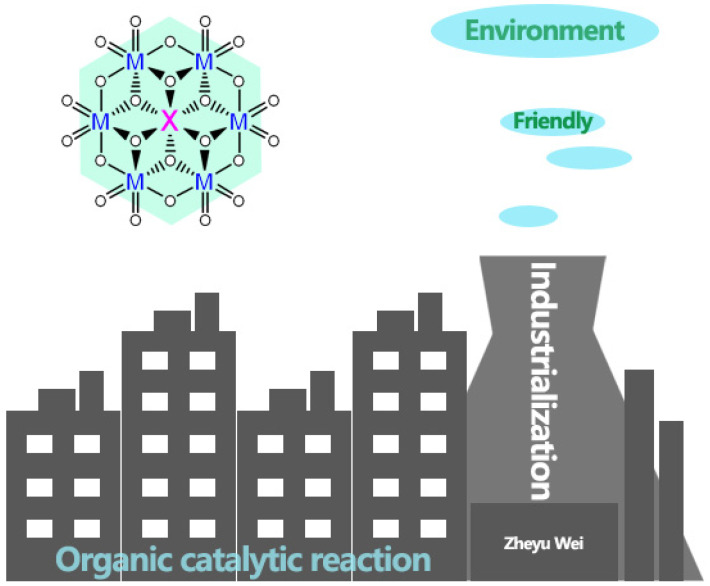
Industrialization expectation of Anderson-type POMs catalyst.

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
