# Peer review of "Recent Advances of Anderson-Type Polyoxometalates as Catalysts Largely for Oxidative Transformations of Organic Molecules"

_molecules, 2022, doi:10.3390/molecules27165212_

Round 1

Reviewer 1 Report

This review aims to provide an update on recent Anderson-type polyoxometalates as catalysts largely for oxidative transformations of organic molecules. I would recommend the title be changed to reflect this rather than the general title. The work that is covered is of a high standard, showcasing catalytic design elements of Anderson POMs towards oxidative processes with a focus on moving towards environmentally friendly conditions. Unfortunately, the review itself is of poor quality. The use of English is poor with many grammatical errors. The figures fail to communicate information clearly and are largely redundant as well as containing mistakes. Citations are absent where necessary and the introduction is often vague and misrepresentative of the catalysis field, thereby requiring more supporting citations. Secondly, some areas misreport data from the papers they are promoting and need to be corrected.

Given the work promoted by this review is of interest and of good quality, this review would do well in Molecules should it be prepared to a higher standard. It may be beneficial to reconsider after major revision. Below are some notes on areas that should be adjusted but this by no means an exhaustive list. Beyond generally improving the quality of the writing, figures that communicate the types of transformation discussed should be included. Largely the work discusses oxidation of amines or alcohols, so it is well within the limits to summarise these nicely in a figure. I would also advise the authors to be less general when addressing limitations in literature as it can come across as naïve. This was particularly apparent in the introduction or when introducing a reaction type; please establish a more specific context of the subject matter, and then discuss limitations as well as some achievements (with citations) to better provide context as to where Anderson POMs fit within the field.

Figure 1 C: the Blue H should be positioned further from the A type diagram for clarity 

Line 42 sentence starting with “their” requires capital.

Lines 54-62 describes crystal growth via new technique to access new Anderson hybrid materials with a citation to a patent. Figure 2 shows some of these crystals. Given the emphasis on this unique technique, figure 2 should include a high quality image of the device in use. Additional citations for the new POM derivatives should be included in this paragraph.

Lines 68-69 Sentence starting with “results show” requires citations for those results. 

Lines 69-72, sentence starting with “In addition,” mentions homogeneous and heterogeneous reactions, a few representative citation examples for both required.

Lines 79-89 establish the need for development of organic synthesis as a global goal, however, the authors refer to catalysis in broad terms saying most need expensive catalysts, harsh conditions, result in by-products. This grossly oversimplifies a complex field. Either specify what areas of organic catalysis is relevant to the paper and its limitations or rephrase this opening to challenges faced when developing catalytic processes. These challenges include reaction economies, accessibility/abundance of catalyst/materials, waste management etc. Suggested readings that may assist authors: Seminal works by Barry Trost on Atom/Step economy towards greener chemistry, Time/Pot economy reviews by Yujiro Hayashi, or ideal synthesis by Paul A. Wender, to name a few.

Lines 84-88 Indicates researchers found that the replacement of traditional metals with Anderson POMs is feasible for a reaction and improved the catalytic efficiency/reduced by-products. What reactions specifically? Citations required.

Line 107, 116, ends in weird symbol, should be full stop.

Line 126, citation at end figure text.

Figure 4. Reaction conditions should be summarised on arrows. According to the paper these conditions are all in one pot and not sequential, thus should not be numbered 1 – 4. Colouring of this figure is misleading. Similar colours indicate connectivity, having R and Cu in pink is redundant. Leave R black and highlight only Cu if desired. Oxygen on the carbonyl is red, O2 black, and OH red, why? Typically, you highlight atoms that are transferring or undergoing change, given O2 is providing the oxygen for the aldehyde oxidation I would suggest O2 and the OH be red and not the carbonyl O. Figure from cited paper is perfect example and should be replicated. https://doi.org/10.1002/cctc.201701599

Lines 137 – 156 discuss a complex mechanism and may benefit from a scheme.

Line 172 remarks at the catalytic system described in citation 27 as being environmentally friendly. This is debatable. Use of benzyl chlorides, and solvents described in the reaction (acetonitrile + work up solvent diethyl ether) score low to moderate for their environmental impact as ranked by GSK. I would remove the sentence starting with “The system is” lines 172-173.

Line 197 ethanol should be changed to represent the general formula, i.e., alcohol 

Figure 5 catalyst 1 to intermediate A should be indicated by two arrows. Currently missing up arrow.

Figure 5 in general is of poor quality. No green on the arrows please. Resolution of image is poor. Unneeded shadow effects on text. The catalytic cycle shown in the cited paper is perfect, replicate this.

Lines 238-241 Discuss catalytic esterification with no citations of existing methods that they claim are so bad. There are many modern academic and industrial processes to esterification that address these issues, authors not representing this reaction fairly.

Line 270 KA oil, KA abbreviation not previously use. Please use full name. alternatively delete “thereof, KA oil”

Figure 6 same issues as figure 4. It also adds very little value to the review and is not essential.

Lines 290 – 293 Sentence starting with “But increasing or decreasing temperature” discusses increased temperature to 70C reduced reaction performance, however, this is not the case. Temperatures in range of 50 – 70 C had yields between 90-99%. It was decreasing temperature to 25C that reduced performance. The rationale for reduced yield also doesn’t line up with temperature discussion. Please refer to original paper for correct data.

Author Response

Dear reviewer:

Thank you for your review. Please see the attachment for specific information.

Zheyu Wei

2022.08.08

Reviewer 2 Report

In this short review concerning Anderson type ([XM6O24]n-) polyoxometalates (POMs) applications in catalysis we can find the recent contributions to this field.

I suggest the authors to increase the quality of images and schemes (molecules are very large compared to the text size).

Author Response

(The authors gave the same response as above.)

Reviewer 3 Report

  The present manuscript reviews recent advances on catalytic organic reactions using Anderson-type polyoxometalates under mild conditions with respect to green chemistry. The Anderson-type polyoxometalate catalysts have been divided into two categories of i) simple Anderson-type anions and ii) organically-modified ones. Several organic reactions mainly on the catalytic oxidation are highlighted with several important references. The topic is interesting and eye-catching as a review article, however, there are critical issues which should be revised. The reviewer cannot recommend this manuscript for publication in the present form. 

1. The referred references are too local, and ca. 1/3 references are written in Chinese. This is not appropriate as a review article in an international journal such as Materials, because these papers are not easy to access for many readers. The authors should refer several previous papers of the group mentioned in the body text (line 50-53). 

2. Figure 2 shows only the photo images of the prepared crystals. Adding molecular names and structures in the figure and figure caption is strongly recommended. The schematic illustration explaining the preparation method of these crystals will be helpful for the readers. 

3. Figure 3 depicts the categories of the Anderson-type catalysts mentioned in the text. However, the relevance of the two categories and the catalysis is not clear. How does the difference in molecular structure (with or without organic moiety) affect the catalysis?

4. The catalysis in the references 22-38 have been summarized in this review. However, there are only two figures accounting these references, which is too short. The authors should add a few more figures or tables.

5. The references of 31 and 33 are not referred in the body text. 

Author Response

(The authors gave the same response as above.)

Round 2

Reviewer 3 Report

The reviewer still wonders if many local references are appropriate for the international journal Materials. However, the manuscript has been greatly improved, and could be published according to the judgement of the editors. 

Minor point: ref. 5 and 10 seems the same. Please check them.